# Fertility correlates with queen size and sperm quality in an ant

**Lena-Marie Süß[1,2], Jan Oettler[1], Luisa Maria Jaimes-Nino[1,3]***

**1** Zoologie/Evolutionsbiologie, Universität Regensburg, Regensburg, Germany, **2** Medizinische Zellbiologie, Universität Regensburg, Regensburg, Germany, **3** Behavioral Ecology and Social Evolution Group, Institute of Organismic and Molecular Evolution, Johannes Gutenberg University, Mainz, Germany

* jaimes.luisa@outlook.com

## Abstract

Fitness can vary between individuals of the same population for reasons related to nutrient acquisition during development, but also to the quality of the mating partners. We focus on ant queens of *Cardiocondyla obscurior* to investigate the effects of female and male morphology and sperm characteristics on productivity. We monitored queens for 12 weeks after egg production started as a proxy for lifetime productivity, and found that larger queens are more productive, as commonly found in other insects, producing more workers and winged males. Sperm viability, but not sperm length, was positively correlated with female-biased sex ratios, pointing to a better insemination rate. The high correlation between sperm viability, male body size, and mandible size suggests strong selection for larger and stronger males. However, male competition for access to unmated queens in the maternal colony may result in a trade-off between male size and developmental time in this species.

## Introduction

Females generally invest a substantial part of their resources in the embryonic development of their offspring, while the male's contribution is typically confined to the quantity and quality of transferred sperm and seminal fluids. In relation to costly female resources in reproduction, a well-documented factor influencing maternal fertility in insects, i.e., demonstrated fecundity, is female body size [1,2]. Adult body size is determined genetically and modified by the quantity or quality of provisioning during immature stages or variation in developmental time [3,4]. In holometabolous insects, adult body size is primarily determined during the larval stage, highlighting the role of the developmental environment in shaping reproductive outcomes [5]. In terms of male fitness traits, body size exhibits a stronger condition-dependency on nutritional limitations during certain life stages than ejaculate traits [6], suggesting that the sperm-related traits are strongly constrained and selected to maintain reproductive function. Generally, nutrient restriction greatly reduces the quantity of seminal

**Data availability statement:** All relevant data are within the paper and its Supporting Information file 2.

**Funding:** JO: OE549/2-2,3 Deutsche Forschungsgemeinschaft https://www.dfg.de/ LMJN: JA3614/1-1 Deutsche Forschungsgemeinschaft https://www.dfg.de/ FAS-G to LMJN Finanzielles Anreizsystem zur Förderung der Gleichstellung of the University of Regensburg https://www.uni-regensburg.de/humanwissenschaften/frauenbeauftragte/frauenfoerderung/index.html No funding sources were involved in study design, data collection and interpretation, or the decision to submit the work for publication.

**Competing interests:** The authors have declared that no competing interests exist.

fluid and sperm, while the quality of sperm, specifically viability and morphology, is less consistently affected. These critical variables appear to be strongly canalized during development, enabling even low-condition males to achieve fertilization [6]. Alternatively, the costs and resource investment for ejaculate traits might be lower compared to those for body size, thereby reducing their dependence on environmental conditions.

In insects in general, where copulation involves a single mating event, the lifetime supply of sperm is retained and maintained in a specialized organ called the spermatheca [7,8]. Spermathecal gland proteins and seminal fluids sustain a high-quality pool of viable spermatozoa by reducing oxidative stress and cellular senescence [9–11]. Further, spermiophagy during long-term sperm storage in ants could possibly eliminate dead sperm cells [12–15], which are a potential source of toxic molecules, enabling a more efficient fertilization of eggs, as suggested for *Melipona* bees [16]. In the social Hymenoptera, male body size is apparently related to greater sperm transfer and mating success in *Pogonomyrmex occidentalis* [17,18], but negatively associated with sperm counts in *Atta colombica* [19]. This suggests different selection mechanisms operating in species where females mate with multiple males.

The ant species *Cardiocondyla obscurior* is characterized by polygynous colonies and intranidal mating of queens with wingless males that have long, sickle-shaped mandibles for killing rivals [20–24]. The wingless male morph maintains lifelong spermatogenesis, allowing them to replenish their sperm supply and mate with multiple females during their relatively long lives, whereas queens typically mate with a single male [21,22,25]. Queen fertility, based on egg productivity, is positively correlated with lifespan [26,27], and longer-lived queens also produce more sexuals [26].

However, variation in lifespan and fertility among queens is high [26,27], and the causes of such variation remain unknown. *C. obscurior* brood is reared in loose piles, with eggs and pupae separated from larvae to prevent cannibalism. Additionally, larvae should experience the same environment and worker care in the nest. Given this presumed lack of variation in extrinsic cues, we sought to investigate other factors influencing differences in offspring number, sex, and caste allocation, and ultimately female fecundity. We hypothesized that maternal condition has a strong effect on the fertility of *Cardiocondyla obscurior* ant queens, while paternal condition does not. However, our findings indicate that queen fertility is correlated with the parental body size of both females and males. Larger queens were also more fertile and produced a more female-biased offspring ratio when mated to larger males with more viable sperm.

## Methods

### Model organism

The *Cardiocondyla obscurior* (Formicidae: Myrmicinae) tramp ant forms small colonies ranging from a few workers up to 150 (median = 28.5, SD = 31.4, n = 62) in their natural habitat [26]. The species originates from Southeast Asia and is widely distributed throughout the tropics and subtropics. It is notable for its male diphenism, with

a wingless ergatoid (from Greek ergat,-'worker' and oid, -'less') male equipped with long, sickle-shaped mandibles, which they employ in lethal fights with rival males to monopolize reproduction, and winged males that are occasionally found [23,28,29]. In contrast to wingless males, the winged male morph completes spermatogenesis upon adult emergence and produces shorter, less variable sperm, though their sperm viability does not differ from that of wingless males [30]. The ants used in this study were collected in 2011 in Japan [31,32]. Since then, the ants were propagated in the laboratory, fed ad libitum with honey, cockroaches, and flies three times per week, and kept in a climate-controlled room on a 12h dark (22°C)/12h light (26°C) cycle, at 75–80% humidity.

## Female and male effects on fertility

Queens mate with a single male and have control over caste and sex allocation [33], making it possible to monitor lifetime investment of single queens [26]. Individual colonies (n = 57) were set up with a dark-colored queen pupa and an adult wingless male of unknown age collected from a different stock colony. Male age is not known to influence sperm quality or offspring production in this species [34,35]. Four freshly eclosed "callow" workers and four L3 last instar worker larvae were added to start the colony, collected from the male's colony to avoid any worker behavior that could be elicited by an alien odor. Once the queen started egg-laying, the larvae were removed, and the number of workers was increased to 15 and subsequently standardized three times per week. Eggs, larvae, workers, and pupae were counted weekly, and pupated individuals were removed three times per week. The queen was monitored for 12 weeks following the onset of egg laying (median onset = 12 days after pupal eclosion, sd ± 3.21). After this 12-week period, the queen was frozen at −20°C. The colony was then monitored weekly until all larvae had eclosed. Overall, the total tracking period extended up to 15 weeks after queen pupal emergence, since the onset of egg laying varied between 9–23 days after mating. Of 57 F1 queens, 42 (73.7%) successfully mated, however, 3 queens died before the end of the observation period of 12 weeks and were not included in the correlation analyses.

## Sperm viability and length

Males from each successful pairing (n = 39) were dissected on a microscope slide in a drop of Beadle solution (128.3 mM NaCl, 4.7 mM KCl, 2.3 mM CaCl$_2$), and the reproductive tissue was transferred to 10 µl fresh Beadle solution. The sperm cells were then released by opening the seminal vesicles and accessory glands (S1A-C Fig in S1 File). Sperm viability was assessed with a LIVE/DEAD sperm viability kit (Molecular Probes, Eugene, Oregon, USA). For staining, 5 µl of SYBR-14 working solution (SYBR stock solution diluted 1:50 in Beadle solution) was added and mixed with the sperm sample. After a 10-minute incubation in a humidity chamber under exclusion of light, 2 µl of propidium iodide was added, mixed in carefully, and incubated in the dark for 7 minutes. Prior to microscopy, a coverslip was placed on the sample droplet. The sperm samples were then examined by fluorescence microscopy (Axiophot, Zeiss, Germany), taking images of 5 randomly selected areas of the sample (20x magnification). For the analysis, the images were blinded, and all live (green) and dead (red) sperm cells of the 5 images were counted, and a proportion was calculated for each male. After determination of sperm viability, measurements for sperm length were assessed by photographing the slides in transmitted light at 40x magnification in 5 randomly selected areas. The length of 10 random sperm cells per picture was measured blindly for each sample using ImageJ [36], tracing a freehand line from head to tail end. For each male, the median sperm length and the standard deviation were calculated across the 5 images.

## Female and male morphometrics

Heads and thoraxes of F0 queens and males were measured using a VHX-5000 digital microscope (20x to 200x objective; Keyence) and analyzed using ImageJ. The morphometric images were blinded before evaluation. For

queens, head width (QHW) and length (QHL), thorax width (QTW) and length (QTL), and spine distance (QSpD) were measured. For wingless males, head width (MHW) and length (MHL), thorax width (MTW) and length (MTL), left and right mandible length (MDL and MDR), petiole width, and antennal segment number were determined. The head was measured in frontal view, length results from the anteriormost point of the clypeus to the posteriormost point of the head, and width represents the distance above the eyes. The thorax was measured in dorsal view, length measured from the mesoscutellum to the end of the mesosoma, width in the queens equals the widest point of the mesoscutellum in front of the wings, and in males at the widest point of the thorax. The distance between the spines was also measured in dorsal view.

## Statistical analyses of female and male effects

The predictor variables used to correlate to the queen's fertility were selected based on their high coefficient of variance and low correlation with other variables (see results section). The correlation matrix for the morphometric data for both queens and males was calculated using Pearson Product-Moment correlation coefficient, *cor.test* (stats R package), for normally distributed data and Kendall rank correlation coefficient Kendall's τ *cor* (stats R package v. 4.4.1) for non-normally distributed data. Significant p-values were obtained using *cor.mtest* (corrplot R package v. 0.95, [37]) and *ggcorrplot* (ggcorrplot R package v. 0.1.4, [38]) for visualization purposes.

Fitted regression models were performed with QTL, median sperm length, and sperm viability as predictor variables. We fit models with gaussian distribution using glmmTMB (R package v. 1.1.10, [39]) to analyze the effect on queen fertility, that is, for the total sum eggs produced, sum of larvae produced, worker pupae, and total pupae counted as

$$\sim \text{QTL} + \text{median\_\mu m} + \text{percentage\_live}$$

When investigating the effect on sexual offspring, the time of sexual production (queen or male pupae production) was included in the model because some queens started producing sexuals at the end of the tracking period of 12 weeks, leaving a shorter time window of data collection. We decided to include the time of sexual production (in weeks) as a continuous covariate when the difference between model AIC was > 2. We modelled the production of queen pupae, ergatoid pupae, winged male pupae, and caste ratio (queens/(queens+workers)) as

$$\sim \text{QTL} + \text{median\_\mu m} + \text{percentage\_live} + \text{time\_sexual\_pupae\_production}$$

with a gaussian distribution using glmmTMB unless specified otherwise. The model to fit the production of winged male pupae used a negative binomial generalized linear model (nbinom2), and the predictor variable "time of sexual production" was dropped because the improvement in AIC was < 2.

Further, we investigated the effect on sex ratio specifying the response as *cbind*(queen pupae, male pupae) and a binomial distribution family (glmmTMB) as

$$\text{sex ratio} \sim \text{QTL} + \text{median\_\mu m} + \text{percentage\_live}$$

and dropping the time of sexual production as covariate because the improvement in AIC was < 2. All models were checked for correct distribution (KS test), dispersion and outliers using the function *simulateResiduals* (DHARMA package version 0.4.6, [40]). Lastly, we tested the effect of maternal size on the male type (winged and wingless males vs only wingless males) and on the sex type (queens and males or queens and wingless males vs only queens) of sexual offspring they produced using a Kruskal-Wallis test (stats R package), and post-hoc pairwise comparison using Dunn's test corrected for a false discovery rate (dunn.test R package v.1.3.6., [41]). All tests and graphs were performed using R (v. 4.4.3 [42]).

## Results

### Higher trait correlation in queens than in males

The body measurements of the queens were all highly significantly correlated to each other (n = 42, τ, p < 0.001; Fig 1A), therefore QTL was used as the representative variable for queen body size in all the regression models (coefficient of variance, $CV_{QHW} = 1.90$, $CV_{QHL} = 2.14$, $CV_{QTW} = 3.00$, $CV_{QTL} = 2.65$, $CV_{QSpD} = 5.41$). The male morphometric data were not as strongly correlated as those of the queens (Fig 1B) and showed greater overall variation (n = 42, $CV_{MHW} = 3.69$, $CV_{MTW} = 4.41$, $CV_{MTL} = 4.47$, $CV_{MDL} = 4.45$, $CV_{MDR} = 3.88$).

### Larger queens are more productive

Queen thorax length (QTL) was positively correlated to the number of eggs (glmmTMB, z = 1.97, p = 0.04), larvae (glmmTMB, z = 2.90, p < 0.01) and the total number of pupae produced (glmmTMB, z = 2.54, p = 0.011, Fig 2A, B). An increase of 1 µm in the queen thorax corresponded to 3.7 (SE ± 1.3) more larvae and 3.3 (SE ± 1.3) more pupae being observed in the span of 12 weeks since egg laying started. The production of worker pupae was also positively correlated with queen size (glmmTMB z = 2.74, p < 0.01; Fig 2C). Similarly, an increase of 1 µm of the queen thorax corresponded to 3.2 (SE ± 1.2) more worker pupae being produced.

The onset of sexual production had a significant effect on the number of wingless males (glmmTMB, z = − 2.75, p < 0.01, Fig 3A), queen pupae produced (glmmTMB, z = − 3.89, p < 0.001, Fig 3B), and the caste ratio produced (glmmTMB, z = − 2.94, p < 0.01, Fig 3C). The earlier queens started sexual production, the more wingless male and queen pupae were produced during the span of the experiment. Therefore, we included the time of sexual production as a covariate when modelling the total pupae production separately by sex and caste. We found that larger queens (higher QTL) produce more winged male pupae (glmmTMB, z = 2.13, p = 0.033; Fig 2D), and a similar number of wingless pupae (glmmTMB, z = 0.84, p = 0.40; Fig 2D). The number of queens produced did not correlate with queen size (glmmTMB, z = −1.08, p = 0.28; S2 Fig in S1 File). Therefore, larger queens have a stronger worker-biased caste ratio (queens/queens+workers,

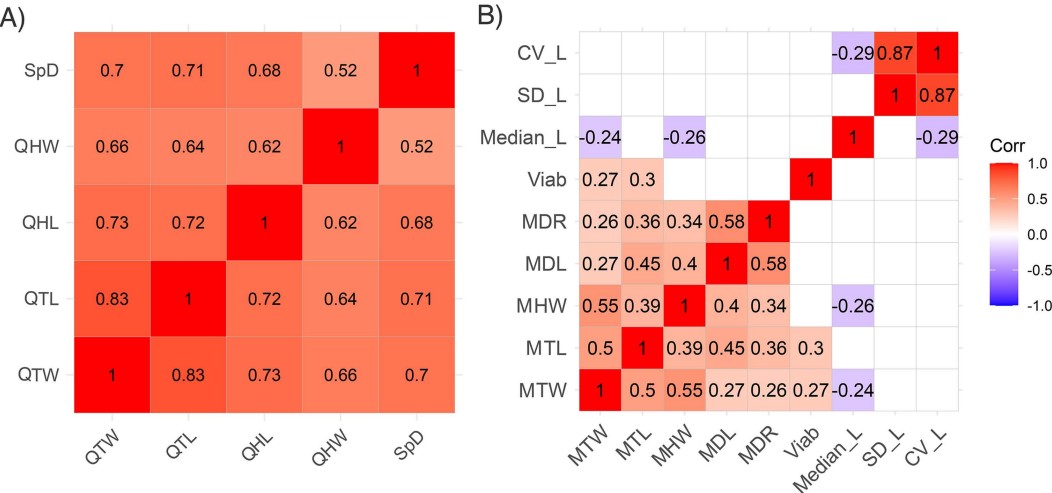

**Fig 1. Morphometric correlation matrix.** Computed Kendall rank correlation coefficients A) queens (n = 42) for thorax width (QTW), thorax length (QTL), head length (QHL), head width (QHW), and spine distance (QSpD). All the performed morphometric correlations were highly significant (τ, p < 0.001). B) Wingless male morphometric correlations (n = 42) for thorax width (MTW), thorax length (MTL), head length (MHL), head width (MHW), mandible length left (MDL) and right (MDR), sperm viability (Viab), as well as the median sperm length (Median_L), standard deviation (SD_L) and coefficient of variance of sperm length (CV_L). The correlation coefficient ranges from a strong negative correlation (r = 1.0, blue) to a strong positive correlation (r = 1.0, red). Only significant correlations (p < 0.05) are shown.

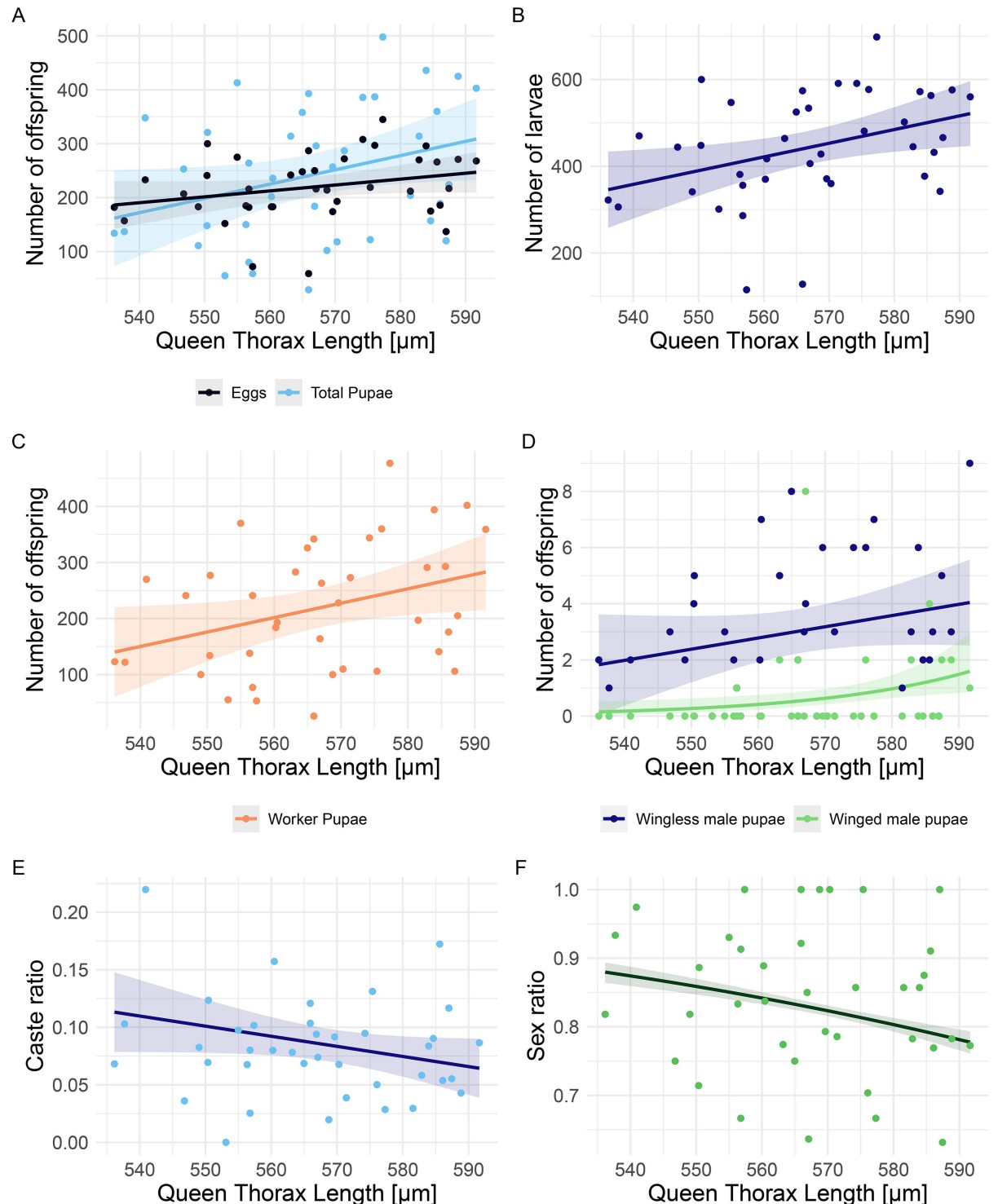

**Fig 2. Correlation of queen body size with fertility (n = 39).** A statistically significant positive correlation was found between queen's thorax length (QTL) and the total A) pupae (light blue), egg (black), B) larvae, and C) worker pupae production. A positive correlation was found between QTL and D) the production of winged male pupae (green), with no significant effect on wingless male pupae ('ergatoid', dark blue) production. No difference in queen production (S2 Fig in S1 File) translates into E) a lower caste ratio (queens/queens+workers), and F) a lower sex ratio (queens/queens+males) for larger queens.

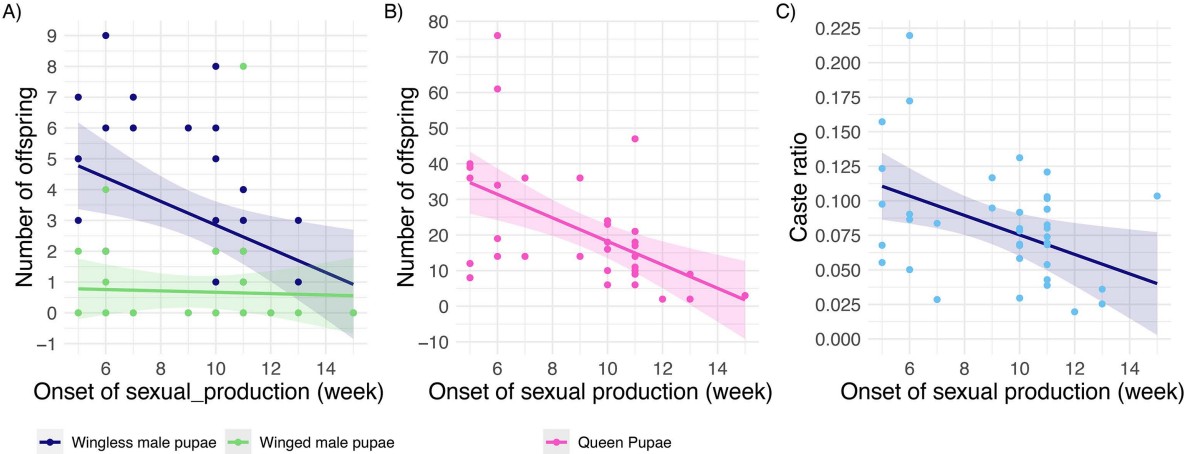

**Fig 3. Onset of sexual production.** Offspring production was monitored 12 weeks after egg laying started, but queens varied in the onset of sexual production. The earlier a queen started producing queen pupae, the more A) wingless male pupae but not winged male pupae were produced. B) The timing of sexual production was correlated with B) queen pupae production and C) the caste ratio (queens/queens+workers).

glmmTMB, z = − 2.41, p = 0.02, Fig 2E), with an increase in 1 µm corresponding to a decrease of 0.1% (SE ± 0.04%) of the caste ratio during the 12 weeks of censoring. Similarly, queen size (QTL) was negatively correlated with sex ratio (queens/queens+males) (glmmTMB, z = −2.37, p = 0.02, Fig 2F). The odds of a pupa being a queen (vs. male) decreases by about 2% with an increase of 1 µm of queen thorax.

QTL was significantly greater in queens producing both ergatoid and winged male pupae (median$_{(Q+E+WM)}$ =579.47 mm, range = 34.85mm) compared to queens producing only ergatoid pupae (median$_{(Q+E)}$ = 560.35mm, range = 49.92mm; Kruskal-Wallis test $\chi_2^2$ = 0.037, p = 0.037; post-hoc Dunn's test z = 2.55, p = 0.033; S3 Fig in S1 File). Queens producing only female sexuals were less fertile than queens that also produced male sexuals. Queens produced a median of 152 eggs when they only produced female offspring, compared to 209.5 eggs when producing queens and wingless males together (Kruskal-Wallis test $\chi_2^2$ = 10.29, p = 0.006; post-hoc Dunn's test z (Q vs. Q + E) = 2.18, p(Q vs. Q + E) = 0.043 and z(Q vs Q + E + WM) = 3.21, p(Q vs Q + E + WM) = 0.004, S4 Fig in S1 File).

**Sperm viability leads to female-biased caste ratios**

A total of 1149 spermatozoa were measured for sperm length, and 24155 sperm cells were checked for viability from 39 wingless males. Sperm viability was 68.4% (cv = 18.6%) on average, which was similar to the average viability when considering only males with > 300 counted sperm cells (mean = 68.8%, cv = 18.4%, n = 27). Sperm viability was positively correlated with sex ratio (queens/queens+males; binomial glmmTMB OR = 12.5, 95% CI [2.62, 59.4], z = 3.17, p < 0.01, Fig 4A). Similar results were obtained when considering all the female offspring (females/females + males; binomial glmmTMB OR = 4.08, 95% CI [1.14, 14.7], z = 2.15, p = 0.031). There was a tendency in males with a higher sperm viability to sire fewer wingless males (glmmTMB, z = −1.91, p = 0.056, Fig 4B).

The sperm quality did not affect other fertility traits as the production of eggs (glmmTMB, median sperm length: z = − 0.91, p = 0.364; viability: z = − 0.99, p = 0.321), winged male pupae (sperm length: z = − 1.09, p = 0.274; viability: z = 0.09, p = 0.932), wingless male pupae (sperm length: z = 0.83, p = 0.404), queen pupae (sperm length: z = − 0.76, p = 0.447; viability: z = 1.06, p = 0.288) or worker pupae (sperm length z = − 1.59, p = 0.113; viability: z = 0.03, p = 0.980). Median sperm length did not affect sex ratio (glmmTMB binomial z = − 0.50, p = 0.620), and the caste ratio was not affected by sperm quality (sperm length, z = 0.25, p = 0.806; viability: z = 1.11, p = 0.27).

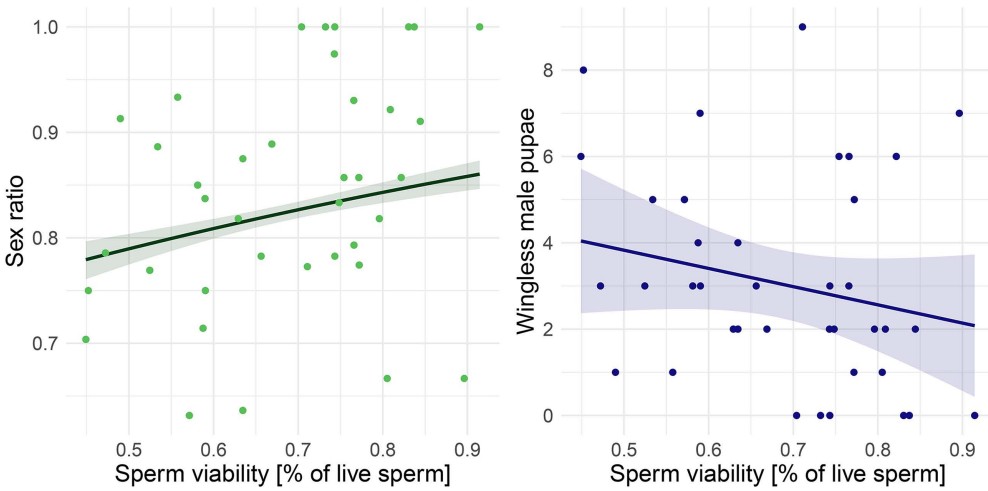

**Fig 4. Influence of male sperm viability on offspring production (n = 39).** A) Higher sperm viability correlated significantly with a stronger female-biased sex ratio (queens/queens+males). B) A tendency was found between the sperm viability and the negative production of wingless males.

## Discussion

As observed in several insect species [2], maternal body size is positively correlated with fertility, i.e., the production of eggs, larvae, and total pupae. Larger queens produced more worker pupae and winged male pupae, but similar numbers of queen pupae during the first 12 weeks after egg laying started. Hence, at a colony level, larger queens invest more into the workforce, while all queens invest equally into the production of female sexual pupae. While 12 weeks is a good predictor of lifetime egg production [43], the production of sexuals reaches a maximum at around 30 weeks [26]. It is possible that larger queens produce more sexuals once the ergonomic phase of the colony is surpassed, i.e., the phase of colony growth with the main investment in the workforce. Still, lifetime egg and sexual offspring production are positively correlated (r = 0.762, and r = 0.691 depending on the study [26,27]), but whether larger queens produce more female sexuals later in life remains to be determined.

While larger queens in polygynous colonies might be responsible for producing most of the workforce, they are the only ones producing winged males. It is possible that the production of winged males is more energetically costly compared to the production of the wingless morph. Most of the males produced in single-queen colonies of *C. obscurior* are wingless (10% winged vs 90% wingless, [26]). Winged male development from egg to adult is one third longer than wingless males (winged 30 days vs 20 days wingless, [44]), and similarly long as queens. Further, the male morphs differ in body size, reproductive tactics, lifespan, sperm length, and accessory gland proteins [28,30,44–46]. Both morphs mate intranidal with closely related females [28], but winged males, considered the dispersal morph, leave the nest later in search of unrelated females [47]. Winged males live a few days to weeks, while wingless males live up to three months [25,48]. Compared to queens, males in *C. obscurior* exhibit greater reproductive variance, as some males will sire a high number of offspring, while many males die in combat and sire none. Theory predicts that parents in good condition should preferentially produce the sex with the higher reproductive variance, whereas lower condition individuals should produce the sex with the lower reproductive variance [49,50]. Larger queens may therefore have more resources for high-risk investment into male production than smaller queens.

In contrast, sperm length in wingless males did not correlate with total fertility, measured by the number of eggs. However, sperm viability was positively associated with queen-biased sex ratios. In social insects, strong selection for high-quality ejaculates is expected due to the prolonged storage of sperm in the spermatheca [8] and the significant immunological costs to females associated with sperm storage [7,9]. Hymenoptera are haplodiploid, with males developing from

unfertilized haploid oocytes. Therefore, ejaculates with more viable sperm will achieve higher insemination rates. Consistent with this, we demonstrated that females mated to males with low sperm viability tended to produce more unfertilized male offspring. Sperm viability is predicted to be higher in species with more intense sperm competition. Evidence from several insect taxa support this hypothesis, as monandrous species tend to exhibit a lower proportion of live sperm [51]. For example, in the seminal vesicles of the polyandrous species *Formica truncorum*, over 90% of the sperm were viable, compared to a median viability of 75% in the monandrous *Dinoponera punctata* [51]. Sperm viability influences paternity success in polyandrous insect species [52]. In *C. obscurior*, females mate with a single male, and male competition occurs predominantly before mating, suggesting a relaxed post-ejaculatory selection on sperm viability that could explain their lower sperm variability (~68%). However, monandrous species not only show lower average sperm viability but also greater variability in sperm quality [51,53], also suggesting lower post-ejaculatory selection. An essay in the polyandrous species *Acromyrmex echinator* and *Atta colombica* reported 60–90% relative sperm viability, along with low variability among males [53]. Exception to this pattern are the monandrous ants *Trachymyrmex* cf. *zeteki* and *Temnothorax crassispinus*, which both exhibited high sperm viability and low variability [53–55], warranting further investigation. Comparison among different studies is difficult given the differences in sperm and seminal fluid across ejaculates, and the different preservation media used.

The wingless male morph is an evolutionary novelty in the genus *Cardiocondyla,* and the ancestral winged male morph has been lost convergently [23]. Unlike males of most ant species, wingless males are relatively long-lived and capable of mating with multiple females [25]. Their spermatogenesis continues during their adult life, allowing for frequent replenishment of the sperm supply [21,30]. Paternal trait correlations showed that larger wingless males had more viable sperm. It is expected that costly sexual traits, such as sperm viability, are less expressed when individuals are in poor condition. Larger males, with a good nutritional status, might be better equipped to maintain the integrity of the germline. In several arthropods and vertebrates, body size responds stronger to changes in nutrition than sperm and ejaculate traits [6]. In line with this, body weight of leaf-cutter ant males correlates with colony conditions [56]. Males are on average heavier than virgin queens at the beginning of the pupal stage, but are lighter at maturity [56], when spermatogenesis is arrested. This could reflect differential feeding rates after eclosion, but direct effects on sperm traits are unknown. Male body size in *Temnothorax crassispinus* show no correlation to sperm viability [55]. Here instead, wingless male sperm viability positively correlated to head and thorax size, and these with mandible length, used to immobilize rivals during combat. Such fights can last for over a day, ending when the stronger male chemically marks the opponent with hindgut secretions, to which workers react and eliminate the defeated rival [20]. However, males with longer mandibles and larger body size may be outcompeted if their developmental time exceeds that of their competitors. Smaller males with faster developmental times may outcompete slower-developing rivals by targeting pupae and newly eclosed males [28]. For instance, in a parasitic wasp, early emergence confers a fighting advantage among males of equal size [57]. Since smaller body size correlates with reduced sperm viability, male size and sperm quality may be under trade-off due to strong sexual selection. Morphological traits exhibited a higher variation in males compared to queens, suggesting variation in the optimal male morphology.

## Conclusion

In summary, during the ergonomic phase of colony growth, colonies with large queens can grow faster due to increased investment in worker production. Larger queens also produce a higher number of winged males that disperse and might contribute additional queen pupae during later phases of colony development, but this needs to be further tested. The size of wingless males, however, appears constrained by opposing selective pressures associated with intra-nidal mating and intense male-male competition. These constraints likely generate divergent evolutionary pressures on male size and ejaculatory traits. We uncovered that larger males achieve higher insemination rates and sire more female offspring. Yet their prolonged developmental time might increase vulnerability to rivals, assuming growth rate remains constant. Together,

these trade-offs underscore the distinctive reproductive biology and evolutionary dynamics of this species. Future studies should focus on how ejaculate traits vary across species differing in mating systems (monandry vs polyandry), and between dispersing and non-dispersing males.

## Supporting information

**S1 File. Supplementary figures.**
(DOCX)

**S2 File. Excel file with raw data and summarized data.**
(XLSX)

**S3 File. R Script to replicate the results of the study.**
(TXT)

## Acknowledgments

We would like to thank Thomas Anthony Keaney for helpful comments. We would like to thank Leonie Riemer for the morphological measurements. We also would like to thank Federico Olivera-Rodriguez, Benjamin Dofka, Melanie Schlossberger, Leonard Mecka, and Maike Fallböhmer for their help in maintaining the colonies. Finally, we want to thank the DFG Research Unit FOR2281.

## Author contributions

**Conceptualization:** Jan Oettler, Luisa Maria Jaimes-Nino.

**Data curation:** Lena-Marie Süß.

**Formal analysis:** Lena-Marie Süß, Luisa Maria Jaimes-Nino.

**Funding acquisition:** Jan Oettler, Luisa Maria Jaimes-Nino.

**Methodology:** Lena-Marie Süß.

**Project administration:** Luisa Maria Jaimes-Nino.

**Supervision:** Luisa Maria Jaimes-Nino.

**Validation:** Luisa Maria Jaimes-Nino.

**Writing – original draft:** Lena-Marie Süß, Jan Oettler, Luisa Maria Jaimes-Nino.

**Writing – review & editing:** Lena-Marie Süß, Jan Oettler, Luisa Maria Jaimes-Nino.

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
