## [Decision Letter · Decision Letter 0]

25 Sep 2025

Dear Dr. Jaimes-Nino,

Thank you for submitting your manuscript to PLOS ONE. After careful consideration, we feel that it has merit but does not fully meet PLOS ONE’s publication criteria as it currently stands. Therefore, we invite you to submit a revised version of the manuscript that addresses the points raised during the review process.

Some questions regarding the experimental design need to be clarified. In particular, you should address the comments made by Reviewer #1 regarding the relatively short monitoring period of 12 weeks and the relatively small number of sperm examined. In the discussion, it would be desirable to place your results in a broader evolutionary context.

We look forward to receiving your revised manuscript.

Kind regards,

Wolfgang Blenau

Academic Editor

PLOS ONE

Journal Requirements:

3. We are unable to open your Supporting Information file “SF3_CODE for manuscript NEW.R”. Please kindly revise as necessary and re-upload.

Reviewers' comments:

Reviewer's Responses to Questions

**Comments to the Author**

1. Is the manuscript technically sound, and do the data support the conclusions?

Reviewer #1: Partly

Reviewer #2: Yes

2. Has the statistical analysis been performed appropriately and rigorously?

Reviewer #1: Yes

Reviewer #2: Yes

3. Have the authors made all data underlying the findings in their manuscript fully available?

Reviewer #1: Yes

Reviewer #2: Yes

4. Is the manuscript presented in an intelligible fashion and written in standard English?

Reviewer #1: Yes

Reviewer #2: Yes

Reviewer #1: General comment

This study investigates how morphological traits of ant queens (Cardiocondyla obscurior) and sperm characteristics of males influence queen fertility, sex ratio, and caste allocation. Using 57 colonies, the authors tracked reproductive output over 12 weeks following the start of egg laying. They show that larger queens produce more workers and winged males but not more queens, while sperm viability (but not sperm length) correlates with female-biased sex ratios. The study highlights interactions between maternal size, paternal sperm quality, and reproductive outcomes, with implications for sexual selection and caste allocation in ants.

The authors make a good job highlighting their research question, the experimental design is overall sound, the statistical analyses are properly implemented, and the results are reported clearly with easy-to-apprehend graphics.

Nonetheless, I have several concerns about the experimental design (and thus the results) that need to be addressed.

Major comments

Comment 1 - Observation period

The monitoring period (12 weeks) is relatively short compared to C. obscurior queen lifespan (~25 weeks). In the study the authors provide for the justification of 12-week productivity as a good proxy for lifetime productivity (reference 41 in the manuscript), the model is indeed statistically significant, but the R2 is low-medium, indicating partial correlation. In addition, this study appears to have been stopped at 15 weeks (not 12), which is close, but should be clearly stated for transparency. In addition, the authors clearly state in the introduction (line 66) that productivity is highly variable between colonies in this species. Moreover, previous studies from the same research group showed that queens shift to the production of sexuals in late life, suggesting that lifetime productivity warrants monitoring over the entire lifespan of queens, because early-stage monitoring can difficulty be extrapolated.

Together, this questions the use of 12-week productivity as a good proxy for lifetime producitivity.

The authors should more explicitly address this limitation and provide rationale as to why the 12-week period was chosen, instead of a longer one such as mean queen lifespan or lifetime monitoring.

Comment 2 - Sperm count

The number of counted sperm is very low (line 255: ‘1149 spermatozoa from 39 wingless males’, which is ~30 counted spermatozoa per male). Sperm viability assay using microscopy are typically performed on several hundreds of spermatozoa per individual (see, for instance, den Boer et al., 2010 – Science; Garcia-Gonzaez & Simmons 2005 – Current Biology; Hunter & Birkhead 2002 – Current Biology) to provide robust estimates of sperm viability.

In addition, sperm viability oscillates between ~0.5-0.9, which reflects a large variance as compared to other social Hymenoptera (see, for instance, den Boer et al., 2010 – Science, Degueldre and Aron, 2023 – Journal of Zoology). The authors rightfully propose that this may be a consequence of low postcopulatory sperm competition and lifelong spermatogenesis. However, the high variance may result from technical issues related to insufficient sperm count.

The authors should address why they chose to limit their analyses to this low number of sperm, and discuss whether the number of counted sperm in their study is sufficient for robust estimates.

If possible, the authors should exploit their photographs to increase the number of counted sperm per male, providing more robust estimates of sperm viability. If not, they should justify their choice to restrict their viability analyses to only a few sperm cells and explicitly acknowledge this limitation in the Discussion.

Comment 3 - Broader evolutionary context

The authors make a good job comparing their result to others in the discussion, however they are mostly limited to their species with a few exceptions. Further contrasting the results from this study in a broader evolutionary context in the final paragraph would add valuable perspective on the generality of the findings and their implications for understanding reproductive strategies across taxa.

Minor Revisions

Lines 52-54: the authors cite articles as reference for the existence of spermiophagy, yet these sources merely speculate that it may happen in the spermatheca, without providing direct evidence. I suggest formulating this sentence with more nuance to the authors’ claim.

Lines 73-74: The use of ‘paternal body size’ is confusing. Do the authors mean ‘parental’?

Lines 75-76: The positive correlation between male body size and sperm quality should be more explicitly explained.

With these revisions, the study would make a valuable contribution to understanding the correlation between inter-individual variations in parental traits and fitness in social insects.

Reviewer #2: General comments

The manuscript titled “Fertility correlates with queen size and sperm quality in an ant” (PONE-D-25-42359) addresses a relevant topic and makes a significant contribution to the field. In my assessment, the text is written in appropriate English and contains original results obtained from well-conducted experiments and analyses. The data support a coherent and pertinent discussion.

Regarding bibliographic references, I recommend that authors omit the months of publication and standardise the format by inserting appropriate spaces between citations. For example, instead of April 1, 1998;42(4):239–46, use 1998; 42(4): 239–46.

Therefore, I recommend accepting the manuscript for publication, but I suggest that the authors carefully evaluate the various suggestions—especially grammatical corrections—indicated in the attached manuscript file.

**Do you want your identity to be public for this peer review?** For information about this choice, including consent withdrawal, please see our Privacy Policy

Reviewer #1: No

Reviewer #2: No

---

## [Decision Letter · Decision Letter 1]

27 Oct 2025

Fertility correlates with queen size and sperm quality in an ant

PONE-D-25-42359R1

Dear Dr. Jaimes-Nino,

We’re pleased to inform you that your manuscript has been judged scientifically suitable for publication and will be formally accepted for publication once it meets all outstanding technical requirements.

Kind regards,

Wolfgang Blenau

Academic Editor

PLOS ONE

Additional Editor Comments (optional):

Reviewers' comments:

Reviewer's Responses to Questions

**Comments to the Author**

Reviewer #1: All comments have been addressed

Reviewer #2: All comments have been addressed

2. Is the manuscript technically sound, and do the data support the conclusions?

Reviewer #1: Yes

Reviewer #2: Yes

3. Has the statistical analysis been performed appropriately and rigorously?

Reviewer #1: Yes

Reviewer #2: Yes

4. Have the authors made all data underlying the findings in their manuscript fully available?

Reviewer #1: Yes

Reviewer #2: Yes

5. Is the manuscript presented in an intelligible fashion and written in standard English?

Reviewer #1: Yes

Reviewer #2: Yes

Reviewer #1: I thank the authors for thoroughly addressing my comments and for providing valuable additional knowledge in the field.

Reviewer #2: (No Response)

**Do you want your identity to be public for this peer review?** For information about this choice, including consent withdrawal, please see our Privacy Policy

Reviewer #1: No

Reviewer #2: **Yes: ** José Lino-Neto

---

## [Editor Report · Acceptance letter]

PONE-D-25-42359R1

PLOS ONE

Dear Dr. Jaimes-Nino,

I'm pleased to inform you that your manuscript has been deemed suitable for publication in PLOS ONE. Congratulations! Your manuscript is now being handed over to our production team.

Kind regards,

on behalf of

Dr. Wolfgang Blenau

Academic Editor

PLOS ONE